# Role of stochastic noise and generalization error in the time propagation of neural-network quantum states

D. Hofmann[1][α], G. Fabiani[2][β], J. H. Mentink[2][γ], G. Carleo[3][δ], M. A. Sentef[1][ε]

**1** Max Planck Institute for the Structure and Dynamics of Matter,
Luruper Chaussee 149, 22761 Hamburg, Germany
**2** Radboud University, Institute for Molecules and Materials,
Heyendaalseweg 135, 6525 AJ Nijmegen, The Netherlands
**3** Institute of Physics, École Polytechnique Fédérale de Lausanne (EPFL),
1015 Lausanne, Switzerland

[α]damian.hofmann@mpsd.mpg.de, [β]g.fabiani@science.ru.nl, [γ]j.mentink@science.ru.nl,
[δ]giuseppe.carleo@epfl.ch, [ε]michael.sentef@mpsd.mpg.de

## Abstract

Neural-network quantum states (NQS) have been shown to be a suitable variational ansatz to simulate out-of-equilibrium dynamics in two-dimensional systems using time-dependent variational Monte Carlo (t-VMC). In particular, stable and accurate time propagation over long time scales has been observed in the square-lattice Heisenberg model using the Restricted Boltzmann machine architecture. However, achieving similar performance in other systems has proven to be more challenging. In this article, we focus on the two-leg Heisenberg ladder driven out of equilibrium by a pulsed excitation as a benchmark system. We demonstrate that unmitigated noise is strongly amplified by the nonlinear equations of motion for the network parameters, which causes numerical instabilities in the time evolution. As a consequence, the achievable accuracy of the simulated dynamics is a result of the interplay between network expressiveness and measures required to remedy these instabilities. We show that stability can be greatly improved by appropriate choice of regularization. This is particularly useful as tuning of the regularization typically imposes no additional computational cost. Inspired by machine learning practice, we propose a validation-set based diagnostic tool to help determining optimal regularization hyperparameters for t-VMC based propagation schemes. For our benchmark, we show that stable and accurate time propagation can be achieved in regimes of sufficiently regularized variational dynamics.

# 1  Introduction

In recent times, the application of machine learning methods to problems in quantum physics has received considerable interest [1]. Examples include the use of neural networks for quantum state reconstruction [2], quantum control [3] and feedback [4], as well as classifying phases of matter [5–7]. Due to their success in approximating high-dimensional nonlinear functions in machine learning applications, neural networks were proposed in 2017 as a variational ansatz for the quantum wave function [8]. These neural-network quantum states (NQS) have been applied to a wide variety of problems in quantum many-body physics, including spin [8–23], bosonic [11, 24] and fermionic [25, 26] systems, as well as quantum computation [27, 28] and dissipative systems [29–32]. One particular research area where NQS could prove important in the near future are non-equilibrium quantum many-body problems, which are of interest across research fields, reaching from quantum simulators with cold atoms and trapped ions [33, 34] via arrays of Rydberg atoms [35], and photonic platforms [36] to laser-driven quantum materials [37]. The theoretical investigation of such scenarios is restricted by a lack of computational methods that allow researchers to reliably simulate driven correlated systems, in particular in two dimensions. NQS provide a promising candidate wave function for the purpose of investigating out-of-equilibrium dynamics, in part due to their ability to capture high-entanglement [38] and topological states of matter [9, 10], which may serve to complement other approaches, in particular those based on tensor network states [39].

    Typically, NQS are time propagated using time-dependent variational Monte Carlo (t-VMC) [8, 40, 41]. So far, this has been studied in the literature primarily in the context of quenches in the spin-1/2 Ising and Heisenberg models in both one and two dimensions [42–47]. In many cases, achieving numerical stability has been identified as the key challenge for the reliable simulation of quantum dynamics [42–44] and also for ground state optimization using imaginary time propagation [19]. In contrast, the capability of the NQS ansatz to represent

the relevant dynamical quantum states was not found to be a limiting factor. However, the general question which types of states can be represented well by a given network architecture and the scaling of the required network size is still a matter of active research [20, 23].

In this work, we are concerned with understanding and separating different sources of instabilities that can prevent t-VMC time propagation to reach dynamical states even when they can in principle be captured by the variational ansatz. To this end, we take a closer look at dynamics in the antiferromagnetic 2D Heisenberg model, which has previously been studied with t-VMC on a 2D square lattice geometry. There, stable time propagation has been demonstrated using the well-established restricted Boltzmann machine (RBM) architecture [45–47]. We compare this setting to the same Hamiltonian on a two-leg ($L{\times}2$) ladder geometry, which features significantly more complex quantum dynamics and which we find to be much more sensitive to numerical instabilities. This is true already for very small systems which are still accessible by exact numerical time evolution (exact diagonalization, ED) which provides us with reliable data to benchmark the RBM dynamics.

We demonstrate that the observed instabilities arise primarily as a result of stochastic error, which is amplified through the generally ill-conditioned variational equations of motion. The stability of the propagation can be improved by reducing noise through means such as increasing the number of Monte Carlo samples or reducing the simulation time step, but this comes at the cost of increasing the required computational resources. However, we show that regularization of the equation of motion also helps to mitigate the effects of noise without significant additional computational cost and highlight the strong effect of regularization parameters on the quality of the resulting trajectory. Taking inspiration from machine learning terminology, this effect can be described as overfitting to stochastic noise, which leads to poor reliability of the time stepping procedure. As in machine learning, this type of overfitting can be detected and quantified by validating the optimized time step on independently sampled data, leading us to introduce a validation-set variational error and show that it can help identify unstable regimes and thus optimize regularization hyperparameters.

The main contributions of this work are therefore (i) presenting the two-leg Heisenberg model as a particularly challenging system to simulate using NQS with t-VMC, making it a useful benchmark case; (ii) the analysis of different sources of error and the effect of regularization on t-VMC propagation in this model; and (iii) the introduction of a validation-set error for quantifying error due to overfitting to stochastic noise.

This article is structured as follows: In Section 2 we define the driven Heisenberg model used as reference in the rest of this paper, in Section 3 we show the influence of regularization on stability and accuracy of the NQS dynamics, and in Section 4 we discuss how this can be quantified using a validation-set approach. Finally, in Section 5, we conclude and provide an outlook on future work.

## 2 Model and methods

We study excitations in the two-dimensional antiferromagnetic (AFM) Heisenberg model (working in units with $\hbar = 1$),

$$\hat{H}(t) = J_0 \sum_{\{i,j\}\in\mathcal{N}} \hat{h}_{ij} + J_0 \, \Delta_x(t) \sum_{\{i,j\}\in\mathcal{N}_x} \hat{h}_{ij} \tag{1}$$

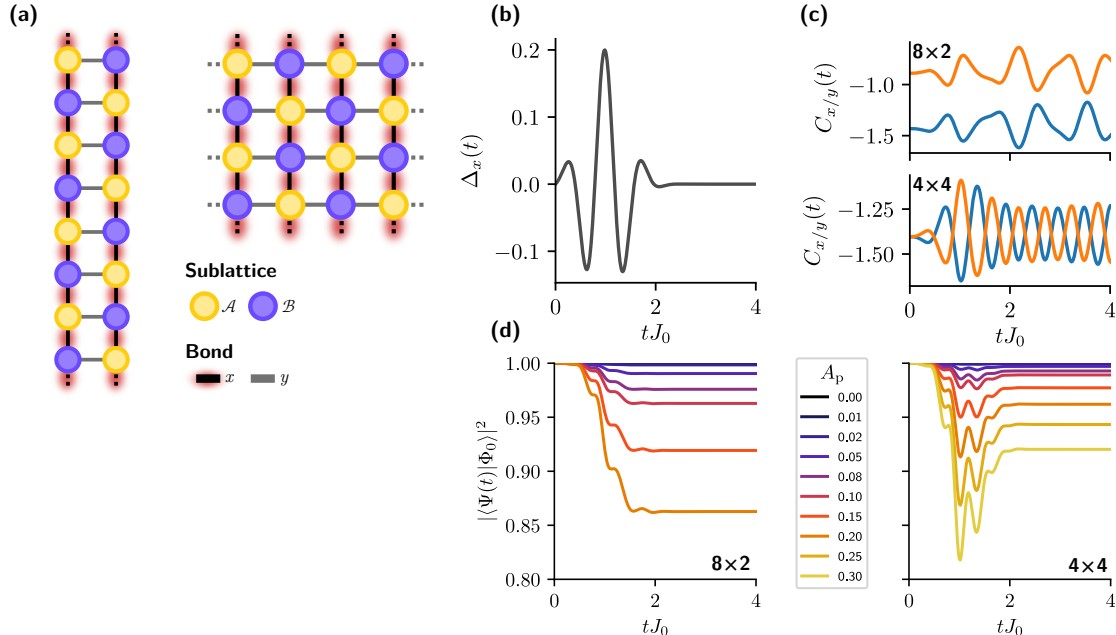

Figure 1:  **(a)** Ladder and square lattice geometry for the 2D AFM Heisenberg model. The colors indicate the $\mathcal{A}$ and $\mathcal{B}$ sublattices of the bipartite model. In the ladder geometry, periodic boundary conditions (PBC) in the $x$ direction are imposed. In the square geometry, PBC are imposed in both directions. **(b)** The Heisenberg system is driven out of its ground state by modulating the $x$-bond coupling in a single pulse with shape $\Delta_x(t)$ [Eq. (3)], here displayed for $A_\mathrm{p} = 0.20$, $t_\mathrm{p} = 0.987 J_0^{-1}$, $\omega_\mathrm{p} = 8.0 J_0$, and $\sigma_\mathrm{p} = 0.4 J_0^{-2}$. **(c)** Oscillations of the $x$ and $y$ bond spin-spin correlations $C_{x/y}(t)$ [Eq. (4)] caused by the pulse for both geometries as computed from the exact time-evolved state. **(d)** Overlap of the initial state $|\Phi_0\rangle$ obtained from ED with the exact time-evolved state $|\Psi(t)\rangle = \hat{U}(t)|\Phi_0\rangle$ for varying pulse amplitude $A_\mathrm{p}$ and both geometries. The other pulse parameters are the same as in panel (b).

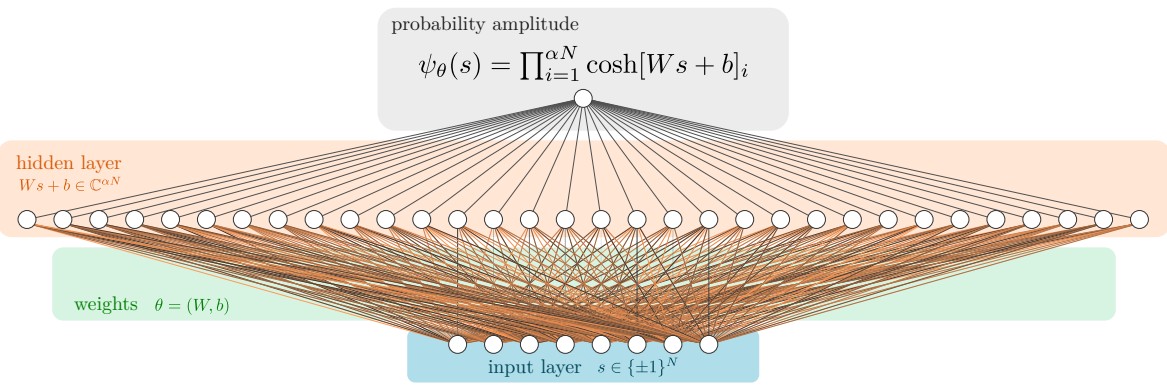

Figure 2:   Restricted Boltzmann machine architecture used as a variational quantum state with $N$ visible units corresponding to the lattice size and a hidden unit density $\alpha$. A detailed description of the ansatz is given in Appendix A.

where

$$\hat{h}_{ij} = \sum_{\mu=1}^{3} \hat{\sigma}_i^\mu \hat{\sigma}_j^\mu, \tag{2}$$

denotes the local Heisenberg coupling acting on each bond with the Pauli matrices $\hat{\sigma}_i^\mu$, $\mu \in \{1,2,3\}$, and exchange coupling strength $J_0 > 0$. Here, the outer sum runs over the nearest-neighbor bonds $\mathcal{N}$ of a finite-dimensional rectangular lattice $\mathfrak{L}$ of size $N = L_x \times L_y$ [Fig. 1(a)] and $\mathcal{N}_{x/y} \subseteq \mathcal{N}$ denotes subset of $x/y$ bonds. We will consider two different lattice geometries: the square lattice with side length $L := L_x = L_y$ and the ladder geometry with $L := L_x$, $L_y = 2$ sites. In both cases, periodic boundary conditions in $x$ direction are assumed. For the square lattice, we also impose periodic boundary conditions in the $y$ direction.

Starting from the ground state at $t = 0$, we study the time evolution of the system under an excitation created by a pulsed modulation of the exchange coupling along the $x$ direction of the lattice (which is the long direction in the ladder system), which has the form [Fig. 1(b)]

$$\Delta_x(t) = A_{\mathrm{p}} \sin(\omega_{\mathrm{p}} t) \exp\left(-\frac{(t - t_{\mathrm{p}})^2}{2\sigma_{\mathrm{p}}}\right) \qquad \text{for } t \geq 0. \tag{3}$$

This driving is physically motivated and can be viewed as a single-cycle THz pulse polarized along $x$ which drives the exchange coupling through a Raman process [45,48]. The $y$ direction coupling is kept constant. In addition to the energy, we compute the average nearest-neighbor bond correlation

$$C_\nu(t) = \frac{1}{N} \sum_{\{i,j\}\in\mathcal{N}_\nu} \sum_{\mu=1}^{3} \langle \hat{\sigma}_i^\mu \hat{\sigma}_j^\mu \rangle \tag{4}$$

along the $\nu = x, y$ direction as an observable. To obtain reference data, we have simulated the time evolution under this pulse through exact (ED) propagation. The resulting dynamics are shown in [Fig. 1(c)]. In both systems, the pulse causes oscillations that persist after the pulse. However, while our driving protocol causes only singlet excitations on the square lattice, the ladder model exhibits both singlet and triplet excitations [49–51] and we indeed

observe more complex and irregular dynamics in the time-dependent bond correlations. For equal amplitude $A_\mathrm{p}$ of the $x$-bond modulation, the ladder system is more strongly affected [as evidenced by the higher distance to the initial state [Fig. 1(d)].

As a variational ansatz, we employ the restricted Boltzmann machine (RBM) with complex-valued weights $\theta \in \mathbb{C}^M$ (Fig. 2) as a parametrization of the quantum wave function $\ln \psi_\theta(s)$ mapping basis spin configurations to the corresponding log-probability amplitudes. The translation symmetries of the lattice are enforced in the manner described in Refs. [8,45], which reduce the number of variational parameters to $M = \alpha(N + 1)$, where $\alpha$ is the hidden unit density. The translation group of an $N = L \times L$ lattice with periodic boundary conditions contains $N$ distinct operations. In the ladder geometry, the notion of periodic boundary conditions only applies to the long direction; however, we include the reflection symmetry along the short direction, so the total order of the translation symmetries is still $N$. The network architecture is fully described in Appendix A.

The time propagation is done using time-dependent variational Monte Carlo (t-VMC) [8,40], which corresponds to numerically solving the equation of motion of the time-dependent variational principle (TDVP)

$$S(\theta(t))\,\dot{\theta} = -\mathrm{i}F(\theta(t), t), \tag{5}$$

where $\dot{\theta} = \mathrm{d}\theta(t)/\mathrm{d}t$, using a stochastic estimate of the quantum Fisher matrix (QFM)

$$S_{ij}(\theta) = \mathbb{E}[\Theta_i^* \Theta_j] - \mathbb{E}[\Theta_i^*]\mathbb{E}[\Theta_j], \tag{6}$$

and energy gradient

$$F_i(\theta, t) = \mathbb{E}[\Theta_i^* \mathfrak{H}(t)] - \mathbb{E}[\Theta_i^*]\mathbb{E}[\mathfrak{H}(t)], \tag{7}$$

with log-probability derivatives $\Theta_i(s) = \partial_i \ln \psi_\theta(s)$ and local energy $\mathfrak{H}(t)(s) = \frac{\langle s|\hat{H}(t)|\psi_\theta\rangle}{\langle s|\psi_\theta\rangle}$. The expectation values $\mathbb{E}[\cdot]$ are taken with respect to the Born probability distribution $\sim |\psi_\theta(\,\cdot\,)|^2$. Further details on the t-VMC propagation scheme are provided in Appendix B. The initial ground state is prepared by minimizing the energy of a randomly initialized RBM using stochastic reconfiguration [41].

# 3 Stability and regularization

In this section, we will highlight jump-like numerical instabilities that we find to arise primarily due to stochastic noise from VMC sampling that enters into the nonlinear equation of motion (5), leading to missteps where the simulation diverges from the physical trajectory in an irrecoverable fashion. We will then show how regularization of the equation of motion can stabilize the dynamics without requiring a change in time step or an increase in Monte Carlo samples.

## 3.1 Numerical instabilities from unmitigated noise

We first highlight the practical challenge posed by the highly nonlinear and stochastic t-VMC equation of motion, by demonstrating how a change in lattice geometry of an otherwise unaltered physical model can affect the stability of the NQS propagation.

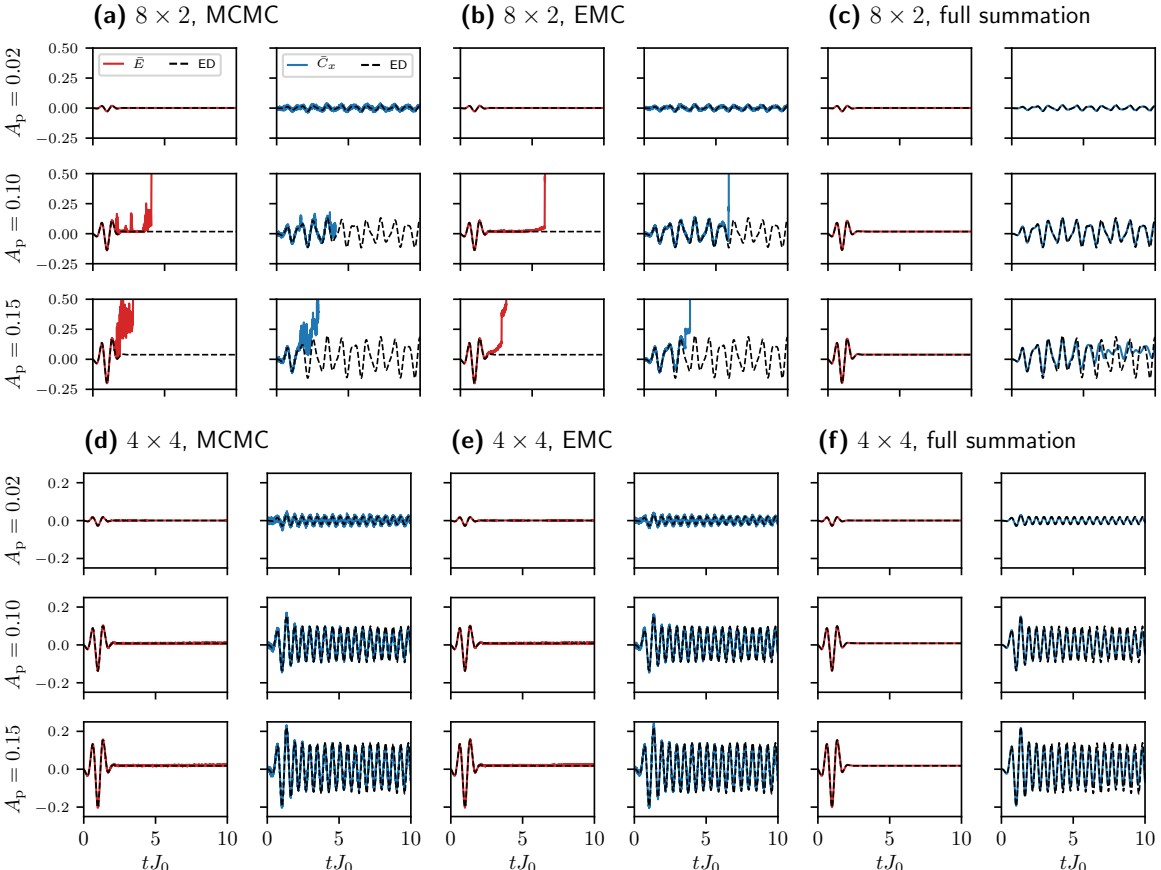

Figure 3: Time-dependent per-site energy change $\bar{E}(t) = [\langle \hat{H}(t) \rangle - E(0)]/N$ and average $x$-bond correlation $\bar{C}_x(t) = C_x(t) - C_x(0)$ for the $8 \times 2$ ladder [panels (a)–(c)] and $4 \times 4$ square geometry [panels (d)–(f)] and varying pulse strengths $A_\mathrm{p}$. The trajectories have been obtained from t-VMC evolution using MCMC [panels (a),(d)] and EMC [panels (b),(e)] sampling as well as results based on full summation of the equations of motion [panels (c),(f)]; see Sect. 3.1 for details. The dashed lines show ED results for reference. In all cases using a symmetric RBM with hidden unit density of $\alpha = 10$ has been used. The initial state is the approximate ground state of the respective system obtained by stochastic reconfiguration and is the same for panels (a)–(c) and (d)–(f), respectively. In all cases, the equation of motion is evaluated by singular-value decomposition of $S$ and applying a diagonal shift of $\epsilon = 10^{-3}$ (see Sect. 3.2 and Appendix D) for regularization.

In previous works [45–47], it has been shown that the dynamics of the Heisenberg model on a square lattice can indeed be successfully simulated using t-VMC with RBM quantum states. We obtain equivalent results for our pulsed driving [Fig. 3(d)–(f)]. The main manifestation of the error as compared to the exact dynamics is a continuous decay of amplitude of the resulting oscillations, which is visible from the averaged nearest-neighbor correlation $C_x(t)$. Increasing the width of the network, i.e., the hidden unit density $\alpha$, both improves the accuracy of the initial (ground) state and reduces the loss of accuracy over the course of the time evolution (data not shown). This shows that the decay is the result of accumulated TDVP error and can be reduced by an increase in network size, which is in agreement with results for a square pulse excitation presented in Ref. [45]. However, this behavior is markedly different for the ladder geometry (with otherwise unchanged system parameters), where instabilities quickly occur during t-VMC evolution already for weak pulse strengths $A_p \geqslant 0.02$ [Fig. 3(a),(b)]. Notably, the observed instabilities violate energy conservation, a property that is inherent to the TDVP equation of motion for a static Hamiltonian. Therefore, their origin has to be numerical. In order to better understand which sources of error in the t-VMC method contribute to these instabilities, we compare three different propagation schemes:

1. t-VMC propagation where the components of the equation of motion are estimated stochastically using Markov chain Monte Carlo (MCMC) with the Metropolis algorithm, which is the standard propagation method for NQS [8,41] [Fig. 3(a),(d)];

2. autocorrelation-free "exact" Monte Carlo (EMC), where samples are directly drawn according to the Born distribution $|\psi(\,\cdot\,)|^2$ without using the Metropolis algorithm [Fig. 3(b),(e)]; and

3. time propagation based on full summation of the t-VMC equation of motion over all spin configurations, which provides a reference free of stochastic noise [Fig. 3(c),(f)].

In the Metropolis MCMC scheme, updates to the spin configuration are proposed based on exchanges of spin pairs which preserve the total magnetization and thus the restriction of the ansatz to the zero-magnetization sector. The EMC scheme can only be applied to small systems accessible to ED, because it relies on the knowledge of the full Born distribution. Here, it is used strictly as a benchmark to uncover the influence of noise on the dynamics while ruling out errors due to non-convergence of the Metropolis sampling[1]. The full summation scheme is similarly limited to small systems with tractable Hilbert space. We have used a second-order Runge-Kutta method (Heun's method) for time propagation in all cases, using two evaluations of the equation of motion (5) per time step, a fixed step size of $\delta t = 0.002$ and $N_s = 7000$ Monte Carlo samples for EMC and t-VMC. Here and in all other MCMC simulations presented in this work, a number of Monte Carlo steps equal to the system size $N$ is performed between each of the $N_s$ samples included in the chain in order to reduce the autocorrelation between successive samples. While there is a visibly increased level of noise with the MCMC sampling, divergences occur at similar time points of the evolution for both approaches. In the full summation results at the same driving strengths, the energy jumps are absent and the dynamics are more accurately reproduced on the ladder with pulses $A_p \leq 0.10$ [Fig. 3(c),(f)]. We note that even in the absence of stochastic noise, the time evolution shown

---

[1]Note that autocorrelation-free Monte Carlo sampling is practically possible beyond the ED regime for NQS architectures based on autoregressive networks [52], though this is quite different from the benchmark implementation considered here.

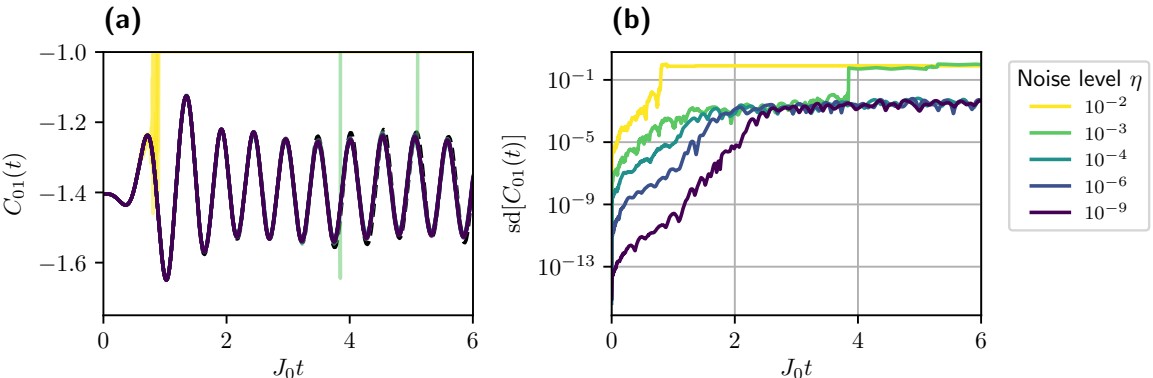

Figure 4: **(a)** Time evolution of the nearest-neighbor coupling term $C_{01} = \sum_{\mu=1}^{3} \langle \hat{\sigma}_0^{\mu} \hat{\sigma}_1^{\mu} \rangle$ using the TDVP equation of motion evaluated by full summation with added artificial noise [Eq. 8]. We show results for the dynamics on the $4 \times 4$ lattice with pulse strength $A_{\mathrm{p}} = 0.20$ for an RBM with hidden unit density $\alpha = 10$. For each value of the noise strength $\eta$, five independent trajectories are shown as faint lines. The opaque lines indicate the median of the respective curves. **(b)** Empirical standard deviation between the set of curves in panel (a) grouped by noise strength $\eta$.

here fails to accurately capture the ladder dynamics for stronger excitations, as can be seen from the $A_{\mathrm{p}} = 0.15$ trajectory. Furthermore, we note that the likelihood of instabilities can be reduced by lowering the integrator time step, which, however, increases the computational cost of the simulation.

In any case, the occurrence of jump-type instabilities is significantly more likely in the presence of Monte Carlo noise. This is true both for MCMC and EMC, showing that the instabilities are not just a result of failed convergence of the Markov chain sampling. Indeed, a noisy energy gradient alone is sufficient to cause the observed divergences when combined with the t-VMC equation of motion for NQS. We show this in an idealized picture as follows: Consider Eq. (5) without any stochastic sampling but with an artificial term of proportional Gaussian white noise added to the energy gradient[2]. This provides a direct way to control the noise level without affecting other steps in the propagation. Specifically, we solve

$$S\dot{\theta} = -\mathrm{i}(F + \xi) \tag{8}$$

where $\xi$ is a random vector with components drawn from a complex normal distribution with mean $\mathbb{E}[\xi_i] = 0$ and variance $\mathrm{Var}[\xi_i] = |\eta F_i|^2$. The parameter $\eta$ determines the relative noise strength and the standard error in each component is proportional to $|F_i|$. The equation of motion is solved using the same second-order Heun scheme used for the t-VMC simulation and at a fixed time step of $\delta t = 0.002$. In Fig. 4(a) we show resulting trajectories for varying noise levels $\eta$. For each value of $\eta$, five independent trajectories are shown. In the absence of instabilities, the standard deviation of the trajectories at first grows with increasing noise

---

[2]We note that the proportional Gaussian noise model of Eq. (8) is indeed a simplification. An analysis in Ref. [44] shows that the actual noise level varies between components of the energy gradient and depends on the quantum geometry of the ansatz (through the QFM spectrum) as well as the energy fluctuations. However, already the simple proportional model used here does exhibit the jump-like instabilities when subjected to noise amplified through the equation of motion.

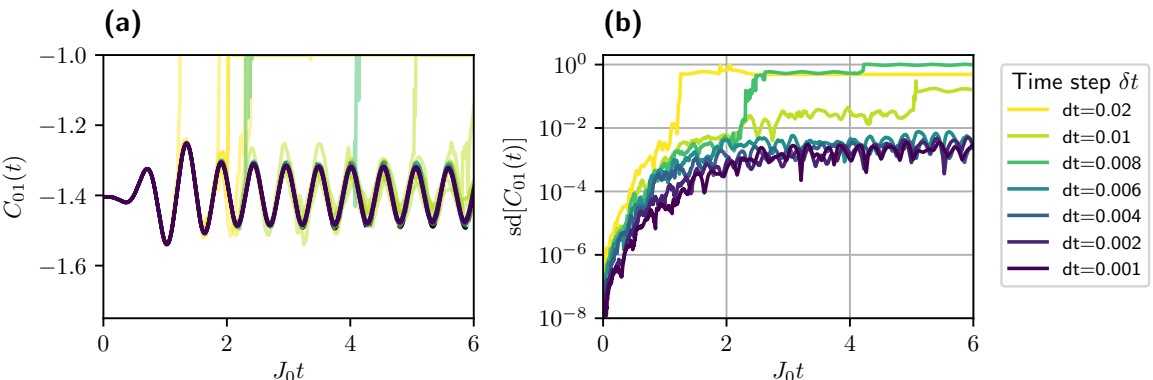

Figure 5: **(a),(b)** Same trajectories as in Fig. 4[(a),(b)] for a driving strength of $A_{\mathrm{p}} = 0.10$ and with varying time step $\delta t$ at a fixed noise level of $\eta = 10^{-3}$.

[Fig. 4(b)]. After a time of the order of the pulse length, the spread of the trajectories stabilizes at a value that is independent of $\eta$. For $\eta \geq 10^{-3}$, jump instabilities occur within the simulation time. Whereas for $\eta = 10^{-2}$ all trajectories show this instability already around $t = 1\,J_0^{-1}$, the jumps happen more sporadically and at later times for $\eta = 10^{-3}$, with only two of the trajectories exhibiting a jump before $t = 6\,J_0^{-1}$. Reducing the integrator time step decreases the frequency of instabilities in a similar fashion, as is shown in Fig. 5.

This idealized experiment shows that random noise in the gradient can be amplified through Eq. 5. Together with the highly non-linear nature of the phase space of the NQS ansatz (compare Ref. [19]), this can cause jump-type instabilities like we have observed in the t-VMC propagation, instead of a gradually increasing spread of the trajectories that would be expected for a more regular ansatz and equation of motion. Both a reduction in noise level and time step reduce the likelihood of instabilities.

In agreement with observations made for other systems [19,44], we find that the expressive capabilities of the network are not a limiting factor. For individual time points, the exact quantum state shown in Fig. 3 can indeed be represented to good accuracy by an RBM of width $\alpha = 10$. See Appendix C for detailed results.

Beyond the data shown here, we have observed RBM states with fewer parameters to be generally more stable. This, however, comes at the cost of decreased accuracy over time, as representational error accumulates (cf. Ref. [45]). The numerical instability present in larger network thus counteracts the benefits of increased expressiveness, making it particularly important to find ways of alleviating this effect without significant increase in computational cost. We further note that while the ladder system is particularly sensitive to the types of instabilities discussed here, they can also occur in the square lattice geometry. We have observed this both in the artificial noise model (Fig. 4) and for a driving strength increased beyond $A_{\mathrm{p}} = 0.30$ (data not shown). Therefore, while the Heisenberg ladder is a very suitable benchmark system for these types of numerical issues, the observations made here can be expected to apply to a broader class of systems, especially when they are driven far out of equilibrium on short time scales.

In the next section, we will discuss how the choice of regularization scheme affects the stability of the dynamics.

## 3.2 Influence of regularization

The formal solution of the TDVP equation (5) is given by

$$\dot{\theta} = -\mathrm{i}S^+ F(t) \tag{9}$$

where $S^+$ denotes the Moore-Penrose pseudoinverse of the QFM [8, 42, 44, 46]. Computing the pseudoinverse can be done by singular-value decomposition (SVD), which is equivalent to the eigendecomposition in this case. This is because $S$ is a covariance matrix and therefore positive semi-definite, i.e., all eigenvalues are nonnegative. Then, in the eigenbasis of $S = V\,\mathrm{diag}(\{\zeta_j\}_{j=1}^M)V^\dagger$, the TDVP equation reduces to

$$\zeta_j[V^\dagger \dot{\theta}]_j = -\mathrm{i}[V^\dagger F]_j. \tag{10}$$

We order the eigenvalues of $S$ by magnitude $\zeta_1 \geq \zeta_2 \ldots \geq \zeta_M$ in the following and denote the smallest nonzero eigenvalue by $\zeta_r$. Then, for all nonzero eigenvalues corresponding to $j \leq r$, we have $[V^\dagger \dot{\theta}]_j = -\mathrm{i}\zeta_j^{-1}[V^\dagger F]_j$. The directions in the null-space of $S$ do not contribute to the physical dynamics; changes of $\theta$ in those directions only affect gauge degrees of freedom of the quantum state. In order to obtain the minimum-norm solution, these are set to zero, i.e., $[V^\dagger \dot{\theta}]_j = 0$ for $j > r$.

In practice, the numerical solution of this equation is complicated by the fact that NQS typically possess a non-exponential but still large number of variational parameters compared to more traditional variational wave functions and further allow for redundancy in the parametrization of a specific quantum state. As a consequence, the QFM is singular, and the nonzero part of its spectrum typically spans many orders of magnitude [43, 44, 53]. Therefore, the linear system (5) has a high condition number $\kappa(S) = \zeta_1/\zeta_r$ which, in particular, means that small perturbations in the right-hand side $F(t)$ can be strongly amplified in the solution $\dot{\theta}$ of the equation of motion (EOM), causing the jump instabilities we have empirically observed above. For this reason, it is necessary to regularize the EOM in order to stabilize the dynamics while preserving the physical accuracy of the resulting trajectory. Typically, this is done by truncating eigenvalues below a threshold $\lambda$ in Eq. (9). Specifically, $\zeta_i$ is treated as zero when $\zeta_i \leq \lambda\zeta_1$. The effective condition number of $S$ is then bounded by $\kappa \leq \lambda^{-1}$. While we focus on this regularization scheme in the remainder, our analysis also applies to a broader class of regularization schemes. In particular, the application of a diagonal shift to $S$ and a signal-to-noise ratio based regularization scheme proposed in Ref. [44] are briefly discussed in Appendix D.

Naturally, there is a trade-off between stability and accuracy: Too much regularization will suppress crucial parts of the physical dynamics, while the system is susceptible to instabilities without or with only weak regularization. This can be seen in Fig. 6, where we show the infidelity of the time-dependent quantum state relative to the ED time evolution of the system,

$$\mathcal{E}(t) = 1 - \frac{|\langle \Psi_{\mathrm{ED}}(t)|\psi_\theta(t)\rangle|^2}{\langle \Psi_{\mathrm{ED}}(t)|\Psi_{\mathrm{ED}}\rangle\langle\psi_\theta(t)|\psi_\theta(t)\rangle}, \tag{11}$$

for varying regularization strength $\lambda$. Specifically for the weak pulse [Fig. 6(a)], we can clearly see a separation of three regimes: an over-regularized regime (for $\lambda \geq 10^{-4}$), where the dynamics are stable but inaccurate; an intermediate stable regime where the physical observables are accurate and the regularization still sufficient to stabilize the dynamics; and an unstable regime ($\lambda \leq 10^{-8}$) where jump instabilities occur within the simulation time

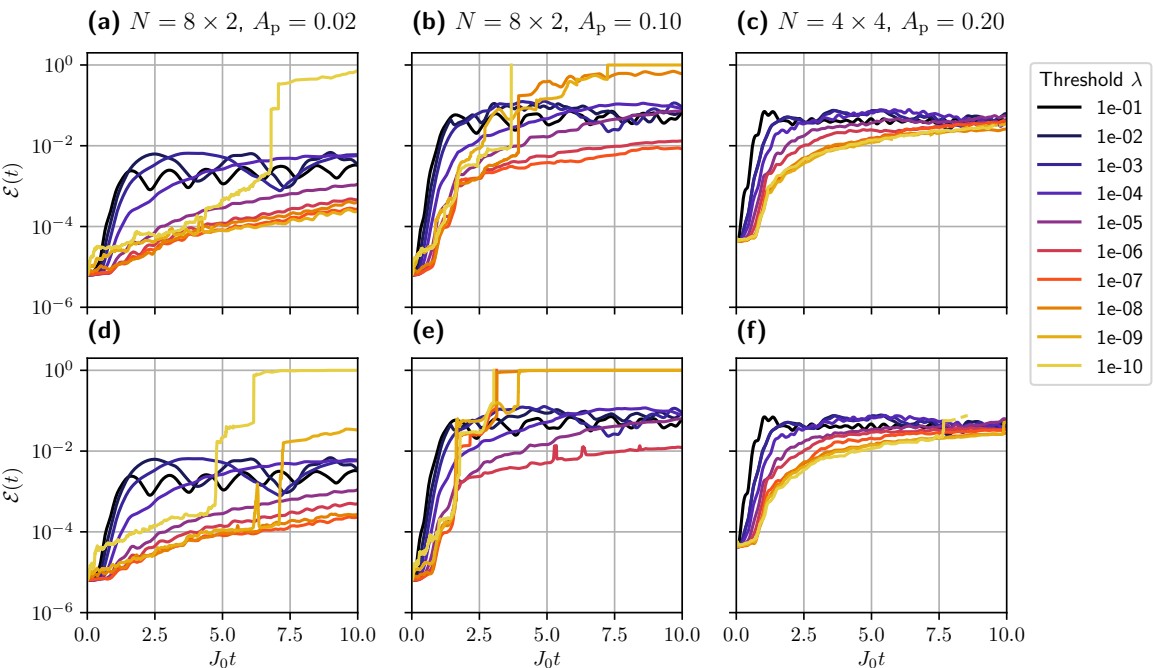

Figure 6: Infidelity of the time-evolved variational state compared to the exact trajectory for varying regularization (in the form of the SVD threshold $\lambda$). The trajectories have been computed using t-VMC with EMC [panels (a)–(c)] and Metropolis [panels (d)–(f)] sampling. The columns correspond to a weak excitation $A_{\mathrm{p}} = 0.02$ [panels (a),(d)] and a moderate excitation $A_{\mathrm{p}} = 0.10$ [panels (b),(e)] in the $8 \times 2$ ladder, as well as a stronger excitation $A_{\mathrm{p}} = 0.20$ in the $4 \times 4$ square lattice [panels (c),(f)]. The initial $\mathcal{E}(t = 0)$ of order $10^{-5}$ is the approximation error of the variational ground state. The time propagation has been computed using t-VMC with EMC sampling with $N_{\mathrm{s}} = 24000$ samples for the ladder [panels (a),(b)] and $N_{\mathrm{s}} = 11200$ for the square lattice [panel (c)]. We have used the symmetrized RBM ansatz with a hidden unit density of $\alpha = 10$.

frame. While for the weak pulse the stable regime spans several orders of magnitude, it becomes smaller with increasing strength [Fig. 6(b)]. By contrast, on the square lattice the dynamics remain stable and largely unaffected by the regularization strength over a wide range of $\lambda$ even at a pulse strength of $A_{\mathrm{p}} = 0.20$ [Fig. 6(c)]. These results have been obtained with EMC sampling, but the same behavior can be observed for the practically relevant case of Metropolis sampling, which is shown in Fig. 6[(d)–(f)]. In this case, the stable regime of regularization becomes smaller for the ladder system. Figure 7 shows the expectation values of energy and bond correlations for several trajectories. These observables show how in the over-regularized regime, numerical stability comes at the cost of physical errors which manifest here in an incorrect reproduction of the oscillation frequencies. At the same time, the square lattice system is almost unaffected by Metropolis sampling, except at very low thresholds. Notably, for both EMC and MCMC sampling strategies, the dynamics converge to a stable trajectory at a much lower number of Monte Carlo samples than in the the ladder system. This hints at an increased sampling complexity of the low-lying ladder excitations in accordance with the higher physical complexity of the ladder excitations, an observation

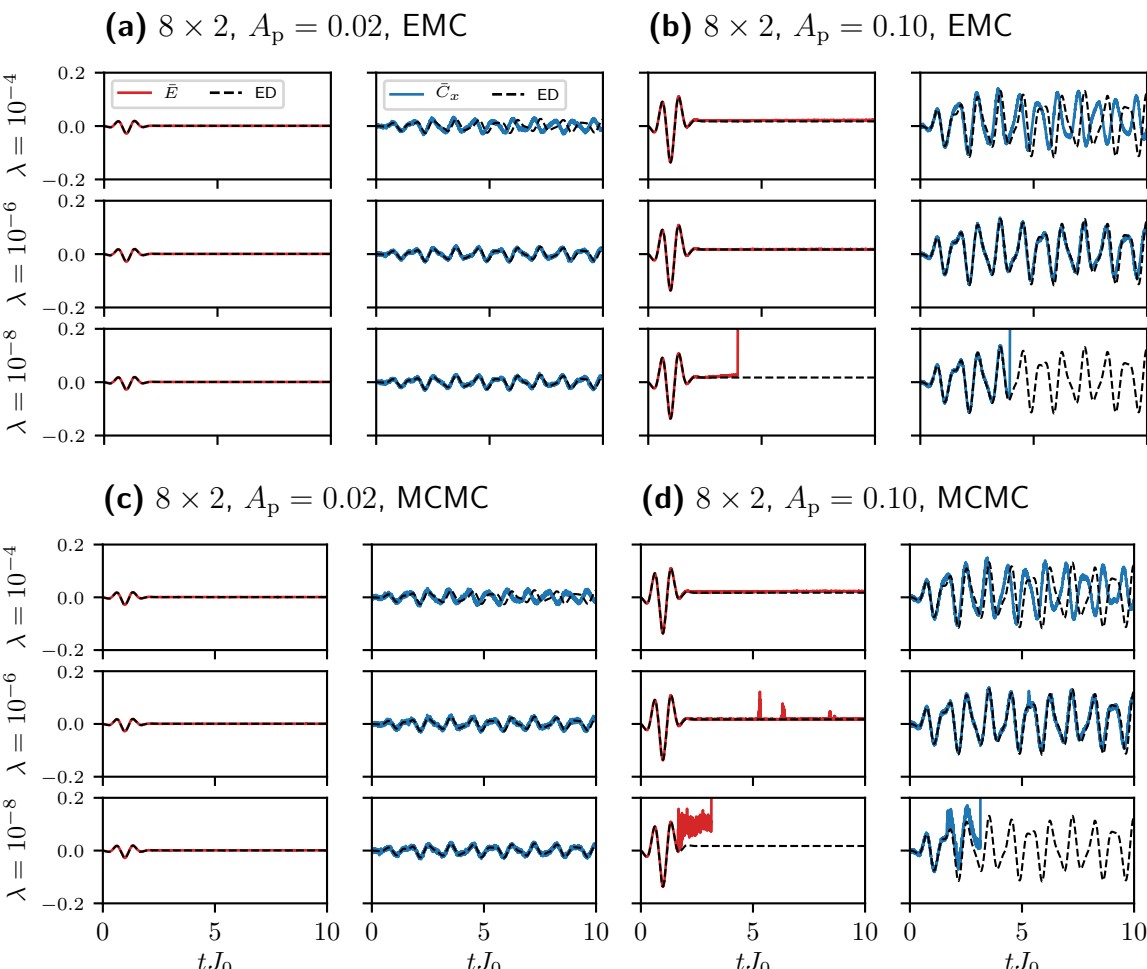

Figure 7: Relative change in energy and $x$-bond correlation as defined in Fig. 3 for a selection of trajectories shown in Fig. 6 with different regularization strengths $\lambda$.

which is corroborated by our results in the next section.

Altogether, our results highlight the delicate balance between stability and accuracy of the dynamics in the presence of stochastic noise and the resulting necessity to fine-tune regularization hyperparameters to reach the optimal regime. From the comparison of ladder and square geometry, we have further seen that the extent of this behavior depends strongly on the details of the system.

# 4 Overfitting to noise and validation error

In order to choose an optimal regularization for a given system and excitation scheme, it is important to have access to appropriate diagnostics. While the error relative to ED as shown in the previous section provides a straightforward way to assess the quality of the solution, this option is restricted to small benchmark systems. Here, we therefore propose an alternative diagnostic which is more generally applicable.

The local truncation error resulting from a single time step in the variational approximation is quantified by the TDVP error [8, 44]

$$r^2(t) = \left[ \frac{D(\psi[\theta(t) + \dot{\theta}\,\delta t], \hat{U}_{t+\delta t, t}\psi[\theta(t)])}{D(\psi[\theta(t)], \hat{U}_{t+\delta t, t}\psi[\theta(t)])} \right]^2. \tag{12}$$

Here, $D(\cdot, \cdot)$ denotes the Fubini-Study distance and $\hat{U}_{t', t}$ is the unitary time evolution operator from $t$ to $t'$. The equation of motion (5) can be derived by locally minimizing the numerator of Eq. (12). The denominator provides a rescaling of the error to account for the varying exact distance between points along the trajectory. This quantity can be estimated to second order in $\delta t$ as [44]

$$r^2(\dot{\theta}; S, F, \delta E) = 1 + \frac{\dot{\theta}^\dagger(S\dot{\theta} + \mathrm{i}F) - \mathrm{i}F^\dagger\dot{\theta}}{(\delta E)^2} \tag{13}$$

where $(\delta E)^2 = \mathrm{Var}[\hat{H}(t)]$ and the other quantities are defined as in Eq. (5).

While capturing loss of accuracy due to the variational approximation, the TDVP error does not account for effects caused by the stochastic noise affecting the equation of motion and thus its solution. In order to account for this additional source of errors, we take inspiration from a standard practice of machine learning: the use of a so called validation error to detect failure to generalize beyond the training data caused by overfitting to a specific sample [54]. Adapted to our present purpose, we consider the specific realization of spin configurations used to estimate the EOM as the training set. Solely optimizing the parameter update for this realization bears the risk of overfitting, in which case the solution may be optimal only on the training set but performs badly on independent estimates of the same EOM. In order to detect this, we can compute two updates $\dot{\theta}^{(i)}$, $i = 1, 2$, from independently drawn samples (separately estimating $S^{(i)}$, $F^{(i)}$ for both). While the resulting $\dot{\theta}^{(i)}$ and corresponding error estimates

$$r^2_{\mathrm{tr},i} = r^2(\dot{\theta}^{(i)}; S^{(i)}, F^{(i)}, \delta E^{(i)}) \tag{14}$$

are identically distributed, the error of the update $\dot{\theta}^{(2)}$ with respect to the independently estimated equation of motion $S^{(1)}\dot{\theta} = -\mathrm{i}F^{(1)}$ can be used to quantify the generalization

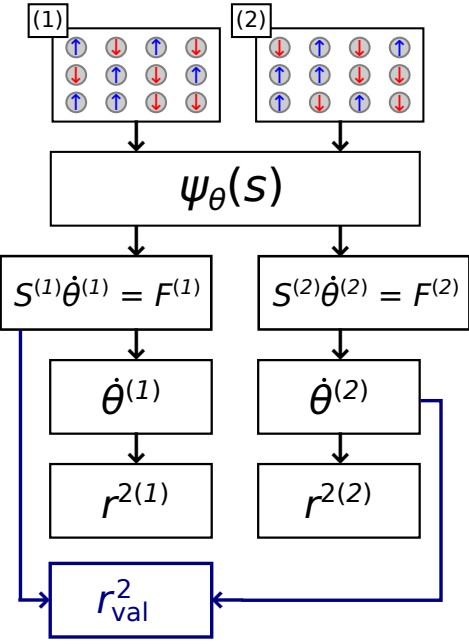

Figure 8: Illustration of the process for computing the validation TDVP error as described in Sect. 4. Two sets of spin configurations are independently generated via Monte Carlo sampling and used to obtain two independent derivatives through solving the equation of motion. The validation error (15) is then computed as the error of the second update with respect to the first equation of motion.

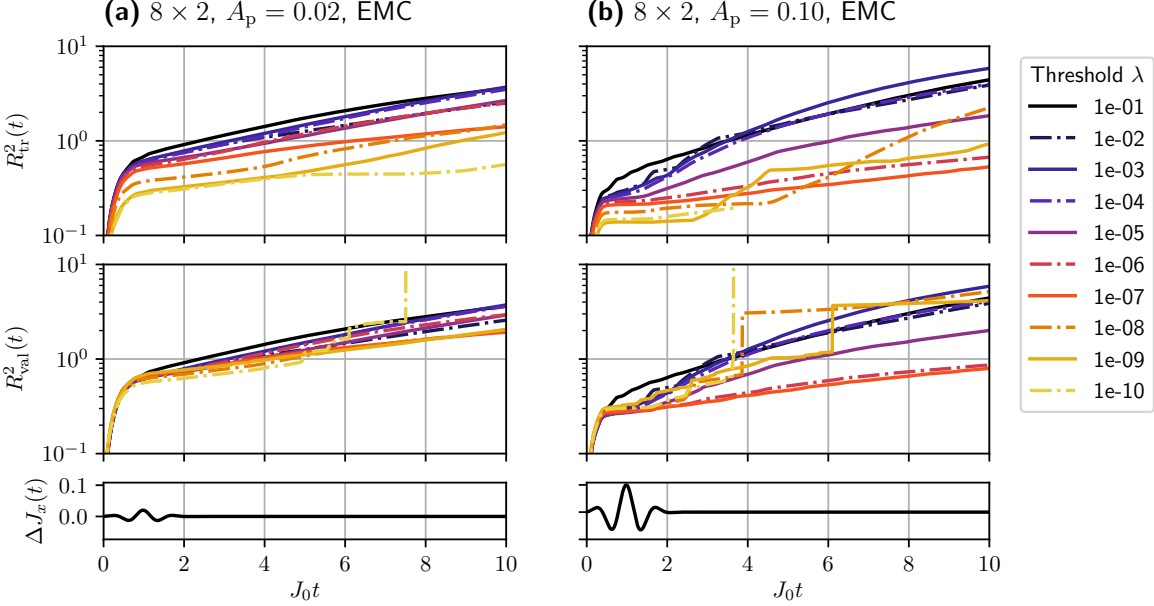

Figure 9: Integrated bare and validation TDVP error (16) over time with varying regularization (SVD threshold $\lambda$) for the same trajectories as shown in Fig. 6(a),(b), i.e., using EMC sampling and with pulse strengths of **(a)** $A_\mathrm{p} = 0.02$ and **(b)** $A_\mathrm{p} = 0.10$. The bottom panels show the time-dependent modulation $\Delta J_x(t)$ for reference.

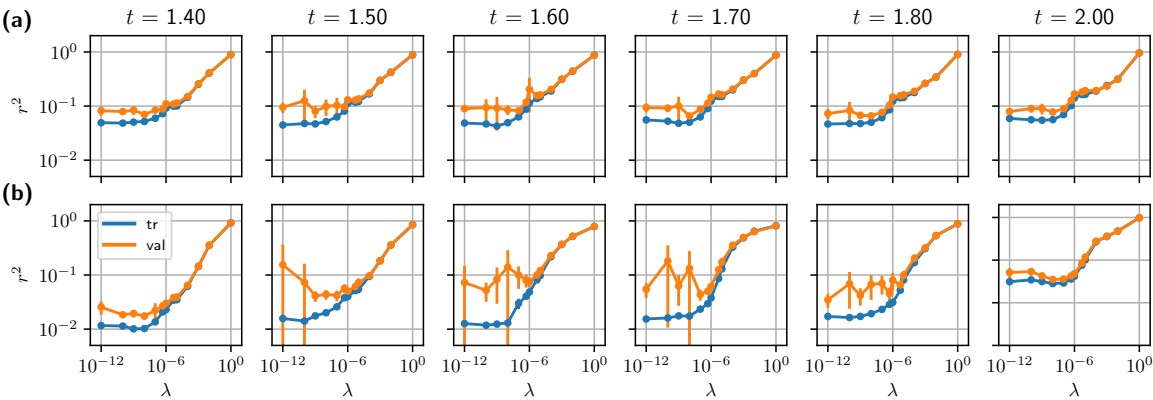

Figure 10: Comparison of the TDVP error $r_{\mathrm{tr}}^2$ and the validation error $r_{\mathrm{val}}^2$ on a logarithmic scale for varying regularization strength in the form of the SVD threshold $\lambda$ at different points in time for **(a)** a weak $A_{\mathrm{p}} = 0.02$ and **(b)** stronger driving amplitude $A_{\mathrm{p}} = 0.10$ on the $8 \times 2$ ladder system. The TDVP error has been computed here by taking variational states $|\psi_{\theta(t)}\rangle$ from the stable $\lambda = 10^{-6}$ trajectory [Fig. 6(d),(e)] and then performing a single step $\delta t = 0.002$ at each displayed time and for each $\lambda$ using Metropolis sampling with $N_s = 28 \cdot 10^3$ samples. We show here the average error over five independent realizations of the validation error, with error bars indicating the standard deviation, in order to account for variance in the error estimate itself.

properties of the parameter derivative. This procedure is illustrated in Fig. 8. Specifically, we define the validation TDVP error as

$$r_{\mathrm{val}}^2 = r^2(\dot{\theta}^{(2)}; S^{(1)}, F^{(1)}, \delta E^{(1)}). \tag{15}$$

Crucially, $r_{\mathrm{val}}^2$ can be estimated using only quantities that are accessible as part of the t-VMC computation. This makes it feasible to use the validation error as a diagnostic for the degree of overfitting and thus reliability of the TDVP solution in systems where a comparison to ED data is no longer possible. If the solution of the EOM is deterministic, we have $\dot{\theta}^{(1)} = \dot{\theta}^{(2)}$ and consequently $r_{\mathrm{val}}^2 = r_{\mathrm{tr},i}^2$. Otherwise, the validation error will be larger, indicating the amount of "overfitting" of the update $\dot{\theta}^{(1)}$ to noise present in the sample. We note that the error estimates are themselves affected by noise in both EOM and energy variance. This is alleviated by considering the integral

$$R_{\mathrm{tr/val}}^2(t) = \int_0^t r_{\mathrm{tr/val}}^2(t') \, \mathrm{d}t' \tag{16}$$

or, if the local quantity is needed, by averaging over additional realizations of $r_{\mathrm{val}}^2$. Despite its ad-hoc nature, we find that this definition of a validation error provides a useful way of quantifying how the regularization scheme affects the solution of the EOM in the presence of noise. Figure 9 shows the integrated TDVP and validation error for weak and moderate driving. While the bare TDVP error is insensitive to the Monte Carlo error and corresponding instabilities[3], a clearly discernible effect is present in the validation error which therefore shows

---

[3]While $R_{\mathrm{tr}}^2$ does increase for small $\lambda$ eventually in Fig. 9, this only happens after the jumps in the corresponding trajectories have already occurred, in contrast to $R_{\mathrm{val}}^2$ which detects hints of the instability already before that point.

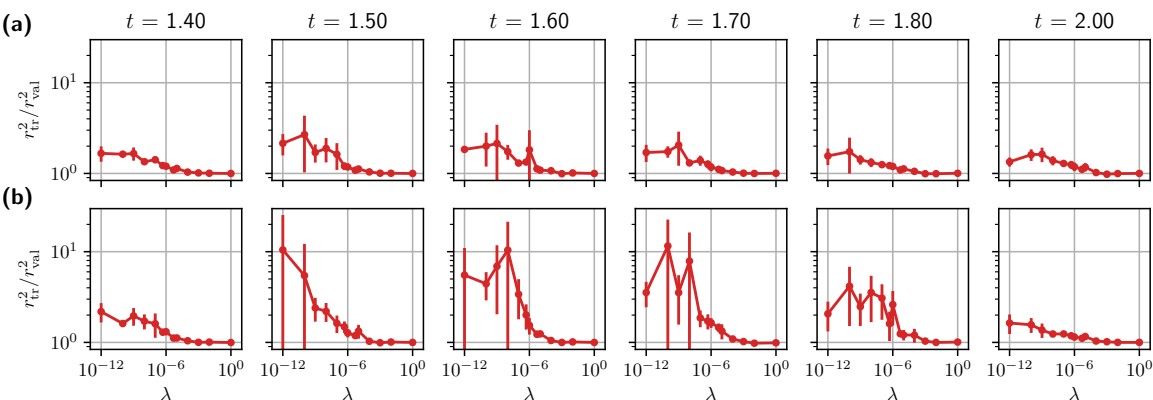

Figure 11:  Validation error relative to the TDVP error for the data shown in Fig. 10.

a much better qualitative agreement with the reference ED error (compare Fig. 6). Note that we show here the error for EMC sampling because, while the utility of the local validation error is not limited to this case, the integrated curves are strongly affected by local perturbances present in the MCMC data.

Figure 10 shows the local TDVP and validation error over a range of thresholds at various times during the duration of the pulse. Here we can see that the unstable regimes of regularization indeed correspond to an increased validation error $r_{\mathrm{val}}^2$ compared to $r_{\mathrm{tr}}^2$, which is consistent with their interpretation as being a consequence of overfitting to noise in the Monte Carlo update, while in the stable regions almost no overfitting error is observed, indicating a high degree of consistency between updates. Furthermore, this behavior is not uniform over time: For $A_{\mathrm{p}} = 0.10$, overfitting occurs particularly strongly at the waning edge of the pulse. The degree of overfitting and its sensitivity to the regularization strength is significantly lower for the weaker excitation. Note that the absolute magnitude of the TDVP error is not directly comparable between different pulse strengths. This is because the denominator of Eq. (12) depends on the distance between $|\psi(t)\rangle$ and $|\psi(t+\delta t)\rangle$ which decreases with decreasing driving strength and goes to zero for vanishing dynamics. Therefore, $r^2$ measures the error relative to the magnitude of the physical dynamics which puts the observed higher values of $r_{\mathrm{tr/val}}^2$ for the weaker excitation strengths into perspective. Data for the validation error relative to the baseline TDVP error can be found in Fig. 11. The validation error also provides some insight into the behavior of the propagation depending on the number of Monte Carlo samples, which we briefly show in Appendix E.

We note that the region of increased overfitting coincides with a region where the exact quantum states, while being representable to a fidelity below $10^{-3}$, appear to be harder to learn using a supervised scheme than states at other times (see Appendix C). However, we would like to stress it is still possible to achieve stable and accurate propagation in this region by suitable tuning of regularization and sample size, showing that an absolute inability of the RBM ansatz to represent those states is not the issue. The precise relationship between the difficulty of supervised optimization, sampling complexity, and generalization error remains an important question for further research, also in comparison with other works [23].

In summary, we have demonstrated that the sensitivity of the ladder system to the regularization scheme as well as the need for a high number of Monte Carlo samples to accurately estimate dynamics especially around the end of the THz pulse is indeed captured by the

proposed TDVP validation error. These effects can thus be seen as a consequence of a lack of generalization of the derivative estimate $\dot{\theta}$ or overfitting to an insufficiently representative sample of spin configurations.

# 5 Conclusions and outlook

We have presented the time propagation of the Heisenberg model on the two-leg ladder as a key benchmark for neural-network-based methods to simulate quantum many-body dynamics. In line with other studies, we have found that RBM quantum states are in principle capable of representing the relevant quantum states during the simulated time evolutions, although important open questions remain regarding the relationship between learnability and sampling complexity of an NQS. However, the combination of (i) numerical instabilities already in small systems and (ii) tunability between relatively well-behaved dynamics on the square lattice and the much more challenging dynamics on the ladder make this model system a suitable case study for t-NQS. Moreover, larger-scale ladders can also be simulated with tensor network states, which makes Heisenberg ladders an ideal *drosophila* for more detailed comparisons between different systematically improvable variational ansätze and propagation schemes beyond system sizes accessible to exact diagonalization.

We have shed light on the delicate balance between stabilizing regularization and physical accuracy of the variational time evolution in the presence of stochastic noise inherent in the t-VMC approach. In particular, motivated by the interpretation of these instabilities as a consequence of overfitting to Monte Carlo noise in the equation of motion, we have introduced a validation-set approach as a quantitative diagnostic of the noise-based error. We have demonstrated that this validation error can be used to aid in the optimization of relevant hyperparameters and can help identifying critical regions where the propagation becomes particularly sensitive to noise. While this is particularly relevant for NQS dynamics, the validation-set approach can be applied to t-VMC simulations using other variational states as well as ground state optimization based on imaginary-time propagation.

The specific validation error introduced here is based on a second-order approximation of the TDVP error, which can be computed from quantities directly available during standard t-VMC runs. However, it is itself susceptible to noise and numerical instabilities. Therefore, while we have shown its capability of quantifiying the influence of regularization and highlighting regions of particularly unstable dynamics, finding a more robust measure of the error may be a useful line of future research.

The ability of quantifying the generalization error in t-VMC propagation also opens up the possibility of devising an adaptive scheme to control regularization hyperparameters and Monte Carlo sampling in order to achieve stable dynamics without the need for manual fine tuning. This is particularly relevant for general NQS software frameworks such as NETKET [55] which strive to be usable in a wide range of physical settings.

## Acknowledgments

We acknowledge helpful discussions with Filippo Vicentini. Simulations have been performed using a t-VMC implementation based on NETKET 2.1b1 [55] as well as code based on ULTRA-

FAST [45]. The exact Schrödinger dynamics have been computed using QUTIP 4.5.0 [56]. We acknowledge support from Flatiron Institute, a division of the Simons Foundation.

## Author contributions

D.H. implemented the t-VMC simulation code based on NETKET, performed simulations and data analysis, wrote the initial manuscript, with contributions from M.A.S., and created the figures. G.F. contributed t-VMC and full summation results using an independent implementation (ULTRAFAST). J.H.M., G.C., and M.A.S. conceived the project. All authors contributed to discussions throughout the project and the preparation of the final manuscript.

## Funding information

G.F. and J.H.M. acknowledge funding from the Shell-NWO/FOM- initiative "Computational sciences for energy research" of Shell and Chemical Sciences, Earth and Life Sciences, Physical Sciences, FOM and STW. M.A.S. acknowledges financial support by Deutsche Forschungsgemeinschaft (German Research Foundation, DFG) under the Emmy Noether program (SE 2558/2).

## A Variational ansatz

In our simulations, we employ the translation-invariant RBM ansatz as introduced in Ref. [8]. Explicitly,

$$\ln \psi_\theta(s) = \sum_{j=1}^{N_{\mathrm{h}}} \ln \cosh\left[\tilde{W}s + \tilde{b}\right]_j. \tag{17}$$

Here, the full weight matrix $\tilde{W} \in \mathbb{C}^{N_{\mathrm{h}} \times N}$ and hidden bias $\tilde{b} \in \mathbb{C}^{N_{\mathrm{h}}}$ are defined in terms of a smaller number of independent parameters $W \in \mathbb{C}^{\alpha \times N}$ and $b \in \mathbb{C}^\alpha$, ensuring that $\psi(\tau(s)) = \psi(s)$ for all $N$ lattice translations $\tau$. We refer to the independent parameters collectively as $\theta = (W, b) \in \mathbb{C}^M$. This ansatz reduces the number of variational parameters to $M = \alpha(N+1)$ where $\alpha = N_{\mathrm{h}}/N$ is the hidden unit density. Thus, the dimension of the parameter space grows only linearly in the system size for the symmetric RBM ansatz. Note that RBM wave functions can include another term, the visible bias $\tilde{a} \in \mathbb{C}^N$, as $\psi_{a,\theta}(s) = e^{\tilde{a}^\top s} \psi_\theta(s)$. When enforcing translation invariance in the manner described above, only one component of the visible bias $\tilde{a}_i = a \in \mathbb{C}$ remains independent which is redundant in the zero magnetization sector and therefore not included in our variational ansatz.

In addition to enforcing translation symmetry, we restrict the state space of the model to the zero magnetization subspace of the full Hilbert space. Thus, the ansatz wave function is only evaluated for spin configurations satisfying $\sum_j s_j = 0$ and $\psi_\theta(s) = 0$ is assumed otherwise. For $N$ sites, the dimension of the full Hilbert space is $2^N$, the zero-magnetization subspace has dimension $\binom{N}{N/2}$. Note that the driving is compatible with both the translation symmetry and zero magnetization constraints. Furthermore, all of our calculations were

performed in a computational basis taking into account the sign structure of the AFM ground state, which is a standard approach for the Heisenberg model [57–59] and helps circumvent the difficulty of learning states with a nontrivial sign structure, which is a more challenging task for NQS [17, 18]. Specifically, this is done as follows: Let $\{|s\rangle \mid s \in \{\pm 1\}^N\}$ denote the $\hat{\sigma}^z$ eigenbasis, so that $\hat{\sigma}_i^z|s\rangle = s_i|s\rangle$. In the AFM phase, the Heisenberg model has a nondegenerate ground state

$$|\Phi_0\rangle = \sum_{s \in \{\pm 1\}^N} \Phi_0(s)|s\rangle \tag{18}$$

which is part of the zero eigenspace of the magnetization $\hat{M}^z = \sum_{i \in \mathfrak{L}} \hat{\sigma}_i^z$. On a bipartite lattice where the sites are partitioned into disjoint subsets $\mathcal{A}$ and $\mathcal{B}$ [compare Fig. 1(a),(b)], the ground state coefficients have the form

$$\Phi_0(s) = (-1)^{\varpi(s)} A_0(s) \tag{19}$$

where $A_0(s) \in \mathbb{R}_{>0}$ is real and nonnegative. The parity $\varpi(s) = \sum_{i \in \mathcal{A}} s_i$ is determined by the magnetization on the $\mathcal{A}$ sublattice. This property is known as Marshall's sign rule [60] and makes it possible to represent the ground state by a real nonnegative wave function in the computational basis $|s^{\mathsf{c}}\rangle = \prod_{i \in \mathcal{A}} \hat{\sigma}_i^z|s\rangle$, which significantly improves convergence of the NQS ground state optimization as the network only needs to learn a trivial sign structure.

## B    Time propagation

In t-VMC [40, 41], the time propagation of the variational ansatz is based on the time-dependent variational principle (TDVP). The equation of motion for the vector of variational parameters $\theta \in \mathbb{C}^M$ is given by $\mathrm{d}\theta(t)/\mathrm{d}t = \dot{\theta}$, where $\dot{\theta}$ is the solution of the linear system Eq. (5). The quantities involved in this equation are the quantum Fisher matrix (QFM)

$$S_{ij}(\theta) = \frac{\langle \partial_i \psi_\theta | \partial_j \psi_\theta \rangle}{\langle \psi_\theta | \psi_\theta \rangle} - \frac{\langle \partial_i \psi_\theta | \psi_\theta \rangle \langle \psi_\theta | \partial_j \psi_\theta \rangle}{\langle \psi_\theta | \psi_\theta \rangle^2} \tag{20}$$

and the energy gradient

$$F_i(\theta, t) = \frac{\partial \langle \hat{H}(t) \rangle}{\partial \theta_i^*} = \frac{\langle \partial_i \psi_\theta | \hat{H}(t) - \langle \hat{H}(t) \rangle | \psi_\theta \rangle}{\langle \psi_\theta | \psi_\theta \rangle}. \tag{21}$$

Here, $\partial_i = \partial/\partial \theta_i$ denotes the complex partial derivative with respect to $\theta_i$. Geometrically, the QFM accounts for the local curvature around $|\psi_\theta\rangle$ on the manifold of variational states. This is analogous to the role of the Fisher information matrix for classical probability distributions [28], which is used in natural gradient descent [61]. The QFM only depends on the form of the variational ansatz and the location in parameter space but not on the Hamiltonian. Still, analysis of its structure and, in particular, its spectrum can give insight into the quantum properties and phase diagram of the system for the RBM ansatz [53].

In t-VMC, both QFM and gradient are estimated as stochastic expectations values $\mathbb{E}[\cdot]$ with respect to the Born probability distribution $\sim |\psi_\theta(\,\cdot\,)|^2$ as written in Eqs. (6) and (7). The resulting equations of motion are valid for a complex differentiable (holomorphic) mapping

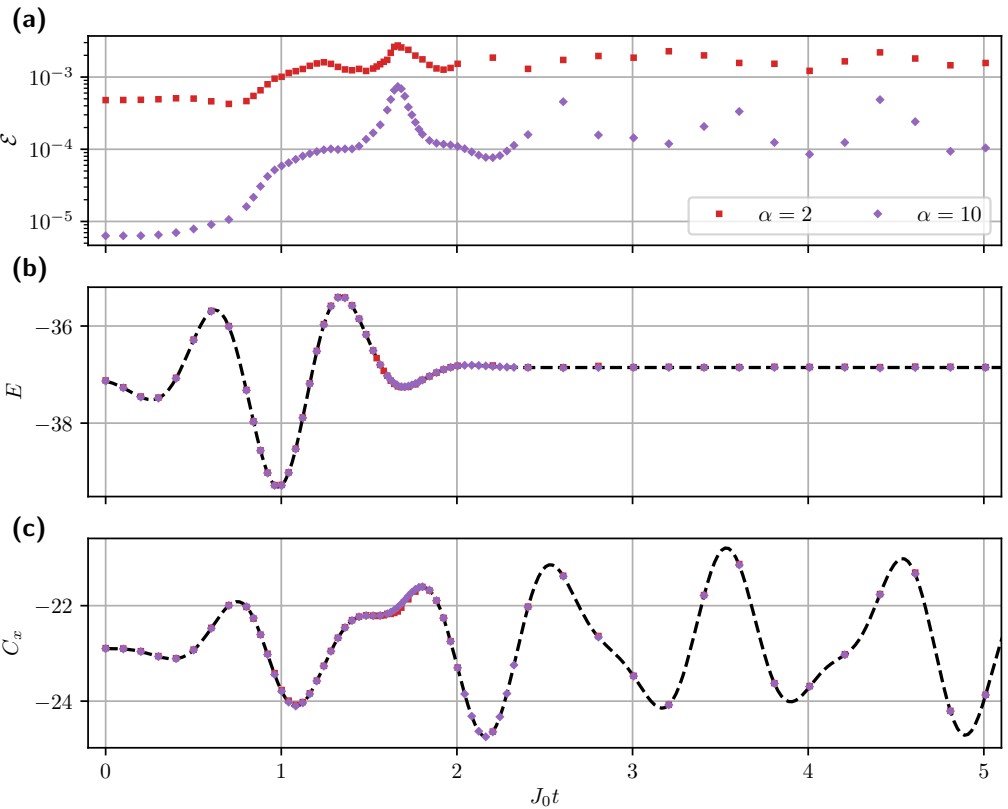

Figure 12: **(a)** Infidelity [Eq. (11)], **(b)** energy, and **(c)** spin-spin correlation $C_x$ for $\alpha = 2$ and $\alpha = 10$ RBM states obtained from supervised learning of the amplitudes along the exact trajectory at an excitation strength of $A_\mathrm{p} = 0.10$. For each time point, the best-fidelity results among five independently optimized states are displayed. See Appendix C for further details.

$\theta \mapsto \psi_\theta$ between the parameters and the quantum wave function. This is indeed satisfied by the symmetric RBM ansatz.

In order to obtain the ground states used as initial states for the time propagation, we have used stochastic reconfiguration [8, 41], which is based on an approximation of the imaginary-time Schrödinger equation in the manner of Eq. (5).

## C    Representability of the trajectory

Even though wider RBMs are necessary in order to better follow the true dynamics via TDVP-based propagation, the states along the trajectories considered here are in general not significantly harder to learn by an RBM than the ground state. In order to test this statement, we fit RBMs with $\alpha = 2$ and $\alpha = 10$ to the exact states along the ED trajectory $|\Psi(t)\rangle$ in the ladder system using a supervised learning approach. The results presented here

have been computed using the supervised learning implementation available in NETKET [55]. Specifically, for each given time $t$, we minimize the negative log-overlap loss

$$L_t(\theta) = -\ln \frac{\langle \psi_\theta | \Psi(t) \rangle \langle \Psi(t) | \psi_\theta \rangle}{\langle \psi_\theta | \psi_\theta \rangle \langle \Psi(t) | \Psi(t) \rangle} \tag{22}$$

using the known probability amplitudes of the exact states that we have obtained from ED. Starting from an approximate ground state, we have run a natural-gradient based optimization [61] targeting this loss function for $n = 1000$ steps at a constant learning rate of $\gamma = 0.01$. This corresponds to a parameter update

$$\theta^{(i+1)} \leftarrow \theta^{(i)} - \gamma \, g_t(\theta^{(i)}) \tag{23}$$

with the loss gradient $g_t(\theta)$ which is the least-squares solution of the linear equation

$$S(\theta) \, g_t(\theta) = \nabla_\theta L_t(\theta). \tag{24}$$

The QFM $S(\theta)$ is defined in the same way as in the main text. In practise, we evaluate the loss $L_t(\theta)$ on batches of spin configurations $\{s^{(i)}\}_{i=1}^B$ of size $B = 1000$ per step, which are randomly drawn from a uniform distribution over all zero-magnetization configurations on the lattice. Note that this update equation is of the same form as in stochastic reconfiguration (SR) [41]. This is because SR is a special case of the natural gradient descent approach applied to the energy expectation value as opposed to a general loss function [28]. The optimization is performed independently for each time $t$. Results for the final infidelity, energy, and spin-spin correlation are shown in Fig. 12. For each time point, the best-fidelity state has been selected from five independent optimizations from the same initial state and with the same parameters[4]. Already at a small hidden unit density of $\alpha = 2$, the trajectory can be represented with an infidelity of the order of $10^{-3}$ with good accuracy in energy and spin correlations. For a wider network at $\alpha = 10$ and starting from a well-converged initial state at $\mathcal{E} = 10^{-5}$, the trajectory can be captured with infidelity below $10^{-3}$ throughout, although a peak of infidelity is clearly visible around $t_* \approx 1.65$. The location of this peak matches the region of increased overfitting and thus instability observed in Sect. 4 (compare Fig. 10). However, even though the peak region is more prone to instabilities, it can still be passed by t-VMC if the time propagation is sufficiently stabilized (compare Fig. 7). Thus, the increased final infidelity of the learned excited states is not necessarily an indication of an absolute inability of the RBM ansatz to capture them more accurately but may also be attributed to an increased difficulty of the optimization.

Altogether, these results provide an upper bound on the minimal infidelity achievable by an optimal RBM representation of the dynamic states at a given size and thus indicate that the representability of the time-evolved states is not the key limitation here. Therefore, an improved time propagation scheme should be expected to be able to reach the accuracy of the supervised learned states.

---

[4]In the region of peak infidelity around $t_*$ defined below, the optimization of the $\alpha = 2$ RBM frequently became unstable after reaching the minimal energy, leading to an increased energy after 1000 steps compared to the actual achievable minimum. Therefore, for the results presented here, the iteration has been stopped early after 10 successive steps without reduction of the loss in this region. The optimization of the $\alpha = 10$ RBM did not have this issue.

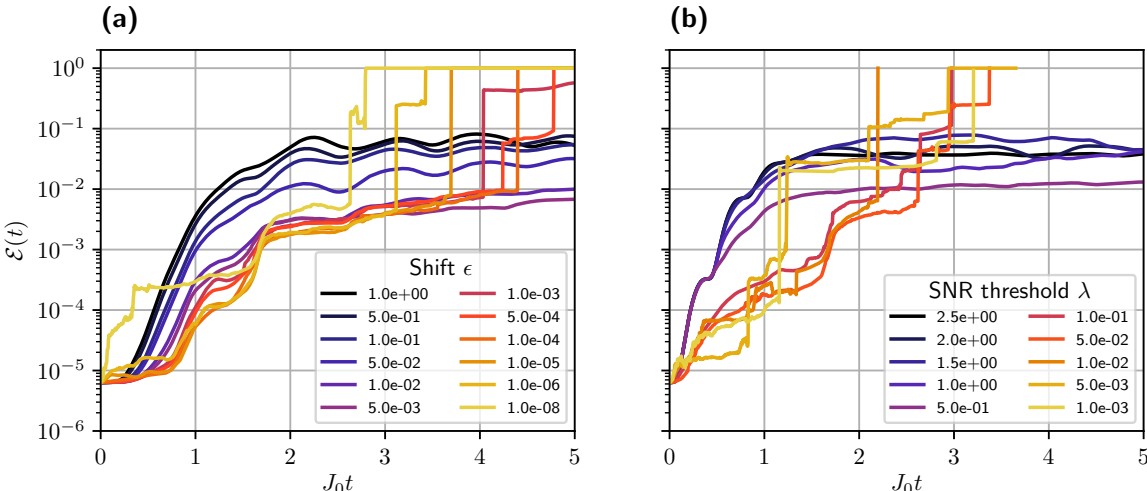

Figure 13: Infidelity of the time-evolved variational state compared to the exact trajectory for **(a)** the diagonal shift regularization with varying strength $\epsilon$ and **(b)** the signal-to-noise (SNR) ratio based regularization of Ref. [44] with varying threshold $\lambda_{\mathrm{SNR}}$. The trajectories have been computed using t-VMC with EMC at a moderate excitation strength of $A_{\mathrm{p}} = 0.10$ in the $8 \times 2$ ladder geometry using the symmetrized RBM ansatz with a hidden unit density of $\alpha = 10$.

## D  Alternative regularization schemes

There are many ways to regularize the linear equation of motion (5) to reduce its susceptibility to noise. We have focused here on the conceptually simple method of truncating the QFM spectrum at a relative threshold as described in Sect. 3.2. Alternatively, the EOM can be regularized by adding a diagonal shift $\tilde{S} = S + \epsilon I$ with $\epsilon > 0$. This is typically done for ground state optimization (see, e.g., Refs. [8, 42, 46]), but can in principle also be applied to the time-dependent case. Since $S$ is Hermitian and positive semi-definite, the spectrum of the shifted $\tilde{S}$ is bounded from below by $\epsilon$, making the matrix invertible and bounding the condition number by $\kappa(\tilde{S}) \leq (\zeta_1 + \epsilon)/\epsilon$. Therefore, a shift significantly larger than machine precision also serves to improve the condition number and stabilize the propagation in a fashion similar to the SVD cutoff. This can be seen in Fig. 13(a) which shows a similar behavior of the shift regularization when compared to the threshold in Fig. 6. A more sophisticated regularization strategy has recently been proposed in Ref. [44]. This approach truncates parts of the equations of motion akin to the singular value threshold above but taking into account the strength of VMC noise in different components of the energy gradient. Specifically, directions in the $S$ eigenbasis are discarded based on a softened cutoff $\lambda_{\mathrm{SNR}}$ of the signal-to-noise of the corresponding component of the energy gradient which can be estimated from the t-VMC data. When applying this approach to the ladder geometry, we have found a behavior similar to the other methods discussed above as a function of varying $\lambda_{\mathrm{SNR}}$ [Fig. 13(b)]. This highlights that the trade-off between stability and physical accuracy we have discussed and the need for reliable diagnostics is relevant beyond the simple regularization scheme used in the main text.

# E    Validation error and Monte Carlo sample size

While sensitive to the regularization, the Monte Carlo error in the update $\dot{\theta}$ is of course also dependent on the number of samples used in the estimate of the equation of motion. The fewer samples are used, the higher the generalization error with respect to the full Hilbert space will be. As with the regularization, this behavior is strongly system and excitation dependent. For the square lattice geometry, convergence with respect to the sample size occurs quickly compared to the ladder system, where a much higher number of samples is needed to obtain reliable estimates. This indicates a higher sampling complexity of the ladder states compared to the well-behaved singlet magnon excitation in the square lattice, as is qualitatively captured by the validation error. This is demonstrated in Fig. 14, which shows the TDVP and validation error, and Fig. 15, which shows the relative validation error, as a function of sample size for both geometries and different driving strengths. We see a clear overfitting behavior which is especially strong around the waning edge of the pulse, as is similarly observed in the regularization dependence in Section 4, and which is only suppressed by increasing the number of samples up to $\sim 28 \cdot 10^3$ for the ladder[5]. In contrast, convergence of the validation error in the square lattice system occurs much faster, even for the stronger excitation shown here.

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
