# Peer review of "Role of stochastic noise and generalization error in the time propagation of neural-network quantum states"

_SciPost Physics_

## Round 2 · Referee Report · Anonymous · 2021-6-25

Strengths

1) High quality in depth analysis on a timely topic, namely the potential of simulating quantum many-body dynamics with variational wavefunctions derived from artificial neural networks.

2) Clarity of presentation and figures.

3) Disentangling of various computational challenges in implementing the time-dependent variational principle with stochastic sampling.

4) A validation error to quantify over-fitting issues is introduced , and its relation to physical observables is demonstrated. While such tools are familiar in the machine learning community, their application to the context of quantum many-body physics is a main merit of this work.

5) The two-leg ladder Heisenberg model is identified as a simple and promising benchmark model for validating the numerical stability of a time-dependent variational ansatz.

Weaknesses

1) Originality claims as well as the relation to previous work could have been worked out more clearly.

2) The high potential of neural network quantum states (NQS) for simulating non-equilibrium dynamics is mentioned repeatedly and prominently. However, the results of this manuscript and previous work on the subject have by no means led to a breakthrough in the computational physics community. Limitations of NQS, as for example reported in Ref. [23] of the manuscript (Lin and Pollmann) are not explicitly mentioned (only cited in bulk).

3) It remains largely open how model-specific the quality of the numerical results is.

Report

See also above strengths and weaknesses.

In their manuscript "Role of stochastic noise and generalization error in the time propagation of neural-network quantum states", Hofmann et al. report on numerical challenges and instabilities in the simulation of non-equilibrium time-evolution of lattice spin models with neural network quantum states. In this context, the authors make an effort to clearly distinguish different sources of errors, ranging from the representational limitations of the chosen variational wavefunction ansatz to the amplification of noise introduced by stochastic sampling in the parameter updates.

In my view two main original directions of this manuscript are: First, a validation error quantifying over-fitting problems in NQS approaches is introduced and linked to errors in physical observables. Second, a simple two-leg ladder Heisenberg spin model is identified as a promising benchmark model for studying the numerical stability of time-dependent variational methods. Compared to comparable 2D models, it does not only have the advantage of revealing more clearly possible instabilities, but its simplicity also allows for better comparison to exact data for reasonable system sizes.

Overall, the presentation is clear and accessible. However, regarding the relation to previous work including more clear originality claims, and a fair assessment of advantages and disadvantages of the NQS approach compared to other computational methods, there is room for improvement.

Requested changes

1) In Eq. (1) it would be better to directly introduce a model with couplings J_\mu, as in the actual simulations the couplings are not only isotropic but partly time-dependent. Hence, it would be better to have as Eq. (1) a model that is closer to what the manuscript is actually concerned with.

2) In Eq. (5), the order of the two expressions should be exchanged, which would make the meaning of "where" more clear. Now, the first expression seems to be a definition of something that occurs in the second one.

3) In the discussion around Eq. (6) it would be great to discuss how the influence of the white noise scales with the size of the time-step.

4) In my understanding \kappa \le \sigma_1/\lambda should read \kappa \le 1/\lambda .

5) It would be better to rename the eigenvalues of the QFM, which are now called sigma, i.e. the same as the Pauli matrices associated with the input spin variables.

  • validity: top
  • significance: good
  • originality: good
  • clarity: high
  • formatting: excellent
  • grammar: excellent

Author:  Damian Hofmann  on 2021-11-19  [id 1957]

(in reply to Report 1 on 2021-06-25)

We thank the Referee for the thoughtful and specific feedback on our manuscript.

Strengths 1) High quality in depth analysis on a timely topic, namely the potential of simulating quantum many-body dynamics with variational wavefunctions derived from artificial neural networks.

2) Clarity of presentation and figures.

3) Disentangling of various computational challenges in implementing the time-dependent variational principle with stochastic sampling.

4) A validation error to quantify over-fitting issues is introduced , and its relation to physical observables is demonstrated. While such tools are familiar in the machine learning community, their application to the context of quantum many-body physics is a main merit of this work.

5) The two-leg ladder Heisenberg model is identified as a simple and promising benchmark model for validating the numerical stability of a time-dependent variational ansatz.

We thank the Referee for highlighting these strengths of our manuscript, which agree well with what we see as the primary aim of our work.

Weaknesses 1) Originality claims as well as the relation to previous work could have been worked out more clearly.

We thank the Referee for pointing out that the relation of our results to previous work and main contributions of this manuscript should be described more clearly.

The main goal of our manuscript is to help the current development in the field of t-NQS methods by identifying a quantitative way to diagnose noise effects and overfitting in the time propagation of NQS, and by proposing the two-leg Heisenberg ladder as a useful test case for improvements in t-NQS methods. Both of these contributions have the goal of aiding in what the Referee has fittingly described as “disentangling of various computational challenges” and to improve our understanding of the strengths and weaknesses of the NQS method for simulating quantum dynamics. We have extended the introductory section in order to clarify these goals and the main contributions of our work accordingly. Furthermore, we have moved some content related to the influence of Monte Carlo sample size from Section 4 to a new appendix section (Appendix F in the revised manuscript) in order to emphasize our focus on the influence of regularization in the main text while retaining additional information that may be useful as a basis for future work in the appendix.

2) The high potential of neural network quantum states (NQS) for simulating non-equilibrium dynamics is mentioned repeatedly and prominently. However, the results of this manuscript and previous work on the subject have by no means led to a breakthrough in the computational physics community. Limitations of NQS, as for example reported in Ref. [23] of the manuscript (Lin and Pollmann) are not explicitly mentioned (only cited in bulk).

We thank the Referee for their feedback regarding our presentation of the current state of research into time-dependent NQS methods. We agree that current limitations of NQS-based approaches are important to investigate. Indeed, the difficulty of stabilizing NQS dynamics, which has been observed in several publications as we discuss in the Introduction, has been our main motivation for studying sources of instability.

Nevertheless, we do consider it justified to classify the NQS approach as “promising”: For a method that was introduced only recently (in 2017), NQS have already been shown to be a versatile tool for exploring equilibrium properties in a range of models, as we mention in the Introduction. It is unsurprising that the nonequilibrium setting is more challenging. However, already the results of this still rather small community (e.g., Refs. [44,45]) do demonstrate a potential which justifies further study. Moreover, NQS have successfully been used to study the physics of a 2D Heisenberg system beyond the lattice sizes accessible by other methods in Ref. [47], which was recently published in PRL.

With regard to the recent preprint [23] mentioned by the Referee, which appeard on arXiv in the month prior to our initial submission, we agree that the question of representative capabilities raised in that work is relevant. Ref. [23] and our work focus on orthogonal aspects: We are concerned with actually reaching states via variational time propagation when we already know they can be learned to a desired accuracy. By contrast, Ref. [23] is concerned with understanding the scaling of required network size to represent certain dynamics. Our analysis in Appendix C shows that the states occuring during our time propagation can indeed be represented to the desired accuracy by the RBM ansatz and, crucially, our required network size exhibits no unbounded growth after the pulse, although there are regions of increased learning difficulty. Other works, both in equilibrium (Ref. [19]) and nonequilibrium (Refs. [44,45]), have also found that representational capabilities of NQS were not a limiting factor in their respective settings.

We have reworded a part of our introduction to more clearly point to the ongoing discussion on representational capabilities and its relation to our work. Furthermore, we have included additional data points for the supervised learning results of Appendix C in order to more clearly show the behavior and help comparisons with other works such as Ref. [23].

3) It remains largely open how model-specific the quality of the numerical results is.

Indeed, we agree with the Referee that the quality of numerical results and, more generally, the performance of the NQS ansatz and t-VMC propagation scheme depends on the model. This can be seen not just from comparing the current literature on NQS dynamics but also directly in our manuscript, where we observe a clear difference between the square lattice and ladder geometries for the antiferromagnetic Heisenberg model. It remains an important question for future work to understand exactly which physical properties of the model determine this complexity and how this is best accounted for in a general t-NQS framework. We hope that by providing the validation error as a tool for diagnosing stochastic effects and by proposing the Heisenberg ladder as a scalable benchmark model, this manuscript can provide a basis for further exploration of these important open questions.

Requested changes

1) In Eq. (1) it would be better to directly introduce a model with couplings J_μ, as in the actual simulations the couplings are not only isotropic but partly time-dependent. Hence, it would be better to have as Eq. (1) a model that is closer to what the manuscript is actually concerned with.

As suggested, we have rearranged the presentation of the model around Eq. (1) to directly introduce the time-dependent Hamiltonian.

2) In Eq. (5), the order of the two expressions should be exchanged, which would make the meaning of “where” more clear. Now, the first expression seems to be a definition of something that occurs in the second one.

We have updated Eq. (5) accordingly.

3) In the discussion around Eq. (6) it would be great to discuss how the influence of the white noise scales with the size of the time-step.

We thank the Referee for this suggestion and have included additional data in Figure 5 of the revised manuscript. This data shows that in the white noise model we have employed, a reduction of the time step does systematically reduce the likelihood of the jump type instabilities. Qualitatively, this behavior is consistent with observations we have made for t-VMC in our present study, where we also find that a reduction in time step size improves the stability of the propagation.

We have expanded parts of the text in Section 3.1 to more carefully discuss this influence of the time step while motivating our decision to focus on the effect and tuning of regularization in the remainder of the study.

4) In my understanding κ ≤ σ₁/λ should read κ ≤ 1/λ.

We thank the Referee for spotting this error, which we have corrected in the revised manuscript.

5) It would be better to rename the eigenvalues of the QFM, which are now called sigma, i.e. the same as the Pauli matrices associated with the input spin variables.

We agree and have updated the text to use a different symbol for the eigenvalues of the QFM.

---

## Round 2 · Referee Report · Anonymous · 2021-7-5

Strengths

1- Detailed analysis of the origin of instabilities of t-VMC with neural-network quantum states.
2- Presentation of a novel diagnostic tool to detect overfitting in t-VMC.
3- Careful separation of the potential sources of numerical issues.
4- Clear presentation.

Weaknesses

1- Although the origin of instabilities was identified, no suggestions on how to overcome the problems are provided.
2- Regularization techniques considered do not include latest developments in the field.

Report

In their manuscript “Role of stochastic noise and generalization error in the time propagation of neural-network quantum states” the authors systematically investigate the origin of numerical instabilities in the simulation of real time dynamics of a quantum spin model. They find that simulations on a ladder geometry are hindered by the occurrence of “jump instabilities” due to overfitting to the stochastic noise that is inherent to the t-VMC method. For their analysis the authors introduce a new diagnostic tool that is based on cross-validation of the solution of the TDVP equation. The study the effect of simulation hyperparameters on the cross-validation error and demonstrate that the overfitting as indicated by the cross-validation error occurs especially for the quenches where t-VMC suffers from instabilities.

This research addresses a timely subject and the manuscript presents valuable insights into what hinders the straightforward application of neural-network quantum states to problems of physical interest. The cross-validation test introduced in this work can help to better illuminate this issue and the identification of the Heisenberg ladder as a “drosophila” provides a suited basis for methodological advancements in future work. Unfortunately, the question whether and how these problems discussed could be resolved remains open, also because the thorough analysis focuses on the simplest regularization techniques and a network architecture that is known to have built-in instabilities.

In general, I find the manuscript suited for publication in SciPost Physics. However, the authors should address the issues listed below.

Requested changes

1- Comparing MCMC to EMC the authors show that remaining autocorrelation in the MCMC simulation seems to result in more severe instability of the algorithm. However, correlated samples can in principle be avoided by increasing the number of proposed updates between two samples in a suited manner or with more sophisticated techniques. Are uncorrelated samples in this case out of reach with MCMC?

2- The maximum number of MC samples used is 2.8e4. Is this the limit of reasonable compute time (e.g., the numbers given in Ref. [44] differ by an order of mangitude)? The data in Fig. 11 give the impression that the overfitting issue might in most cases vanish around 100k samples. It would be helpful to add a comment about the feasibility of simulations with more samples.

3- In Appendix C it is not totally clear to me which optimizer is being used. The authors mention “stochastic reconfiguration”, but it is not clear to me how this is used for supervised learning. Also with a potential comparison to Ref. [23] in mind it could be worth to better clarify this point.

4- Below Eq. 5 it would be helpful to give the formulas for the quantum Fisher matrix and the energy gradient.

5- On page 6 the authors discuss the step size used and mention that smaller step sizes were tested. What was the smallest step size tested and why wasn’t an integrator with adaptive step size used?

6- Axis labels in Fig. 9 seem misleading. The labels seem to denote a ratio, but after thinking about it for a while I believe that the “/“ is rather supposed to denote an “or”. This should be fixed.

7- The authors refer to the quantity "1-fidelity" as the “inverse fidelity”. I think the more commonly used term is “infidelity”. The authors might consider to change this.

  • validity: high
  • significance: good
  • originality: high
  • clarity: high
  • formatting: excellent
  • grammar: excellent

Author:  Damian Hofmann  on 2021-11-19  [id 1958]

(in reply to Report 2 on 2021-07-05)

We thank the Referee for the thoughtful and specific feedback on our manuscript.

Strengths 1- Detailed analysis of the origin of instabilities of t-VMC with neural-network quantum states. 2- Presentation of a novel diagnostic tool to detect overfitting in t-VMC. 3- Careful separation of the potential sources of numerical issues. 4- Clear presentation.

We thank the Referee for pointing out the detailed analysis of the instabilities and the proposal of the validation TDVP error as a novel diagnostic tool for noise-sensitivity and overfitting in t-VMC propagation as strengths, which we see as main contributions of this work.

Weaknesses 1- Although the origin of instabilities was identified, no suggestions on how to overcome the problems are provided.

We respectfully disagree with the Referee that “no suggestions are provided”. Based on our data, we demonstrate that instabilities can be mitigated and at least to an extent overcome by careful tuning of the employed regularization and, as far as computationally feasible, choosing a larger amount of MC samples (which can differ strongly between different models of the same size, as we show through our comparison of the square and ladder geometries). With our noise-sensitive validation error we provide a tool to help diagnose the stochastic error introduced by t-VMC and thus aid in performing this tuning of simulation parameters. Given the Referee’s feedback, we conclude that these points should be highlighted more clearly, which we have done in the Introduction and Conclusion sections of our revised manuscript.

While it is true that this alone does not resolve the issues of the instabilities in all scenarios (nor do we claim to address all potential sources of numerical instability in this work), we do think our analysis and suggested diagnostics are helpful to researchers encountering these problems in practice and can serve as a basis for future efforts towards achieving stable NQS time propagation across a wider range of scenarios.

2- Regularization techniques considered do not include latest developments in the field.

While we focus on the conceptually simple regularization scheme based on a cutoff of low-lying singular values in the main text, we have also included data on a regularization using a diagonal shift and, in particular, a recently proposed regularization method based on the signal-to-noise ratio of individual gradient components (Ref. [44]) in Appendix C. To our knowledge, these are the main methods used for time-dependent NQS studies at this time and, as we demonstrate in our Appendix D, all of these methods share the need for careful tuning of a cutoff or similar hyperparameter in order to find an optimal balance between mitigating noise and suppressing physically relevant dynamics, so that our analysis and suggested diagnostic are relevant in all these cases. Therefore, we consider our work relevant in relation to the current state-of-the art in NQS time propagation.

[...] Requested changes

1- Comparing MCMC to EMC the authors show that remaining autocorrelation in the MCMC simulation seems to result in more severe instability of the algorithm. However, correlated samples can in principle be avoided by increasing the number of proposed updates between two samples in a suited manner or with more sophisticated techniques. Are uncorrelated samples in this case out of reach with MCMC?

First we would like to clarify that we do already reduce MCMC autocorrelation by thinning, i.e., by performing several proposed updates (equal to the system size) between samples included in the chain and have added this information to the revised manuscript. Generally, thinning of the chain reduces autocorrelation at the cost of using a smaller number of samples overall, so these two contributions can counteract each other. Further thinning of the chain is not necessarily preferable to simply increasing the sample size for this reason.

In the absence of more severe convergence failures of the Metropolis sampling, we thus expect the MCMC results to match EMC results at a lower sample size, which is consistent with our results. Importantly, our results suffice for establishing that jump instabilities cannot be attributed solely to failed MCMC convergence due to their occurrence in EMC results, which is our main motivation for including the later in our manuscript.

2- The maximum number of MC samples used is 2.8e4. Is this the limit of reasonable compute time (e.g., the numbers given in Ref. [44] differ by an order of magnitude)? The data in Fig. 11 give the impression that the overfitting issue might in most cases vanish around 100k samples. It would be helpful to add a comment about the feasibility of simulations with more samples.

As the Referee points out, it is indeed feasible to perform NQS simulations using of the order of 10⁵ samples as it was done, e.g., in Ref. [44]. We note, however, that in our simulations of the Heisenberg model we can work in the zero-magnetization sector, which is easier to efficiently explore using MC sampling than the full Hilbert space as has to be done, e.g., for the Ising model (which is studied in Ref. [44]). This has to be taken into account when comparing sample sizes between our work and Ref. [44] or similar studies. Moreover, for the square lattice Heisenberg model we have observed a significantly faster convergence of the dynamics, making the required number of samples to achieve a comparable error in the Heisenberg ladder quite remarkable. Thus, while technically feasible, sample sizes of the order of 10⁵ (an order of magnitude larger than the Hilbert space dimension) appear excessive for the benchmark system under consideration and we therefore prefer to focus on the behavior and potential of stabilizing the dynamics through regularization in the regime of lower sample sizes. We have added a comment in this regard to the paragraph discussing Fig. 11 to the text.

3- In Appendix C it is not totally clear to me which optimizer is being used. The authors mention “stochastic reconfiguration”, but it is not clear to me how this is used for supervised learning. Also with a potential comparison to Ref. [23] in mind it could be worth to better clarify this point.

We thank the Referee for pointing out that our description of the supervised learning scheme can be clarified. What we have referred to as “stochastic reconfiguration” in this context should more precisely be called natural gradient descent (NGD) applied to the loss function of Eq. (22), which amounts to a rescaling of the loss gradient by the pseudoinverse of the QFM. This method is precisely stochastic reconfiguration when the loss is the energy expectation value, but the principle of NGD can be applied to general loss functions as we have done here for the negative log-overlap (NLO) loss. We have clarified this in the text of Appendix C.

The NLO loss function we have used has been proposed in the context of NQS in Ref. [27] and is also discussed as one of several options in Appendix C of Ref. [23], though it is not the one used to obtain the main results of that preprint. We have used this specific form of the loss because it can handle unnormalized probability amplitudes, which is beneficial for use with our NQS implementation, and because of its direct availability in NetKet. The results are sufficient for our purpose, which is to show that the states occuring along the ED trajectory of our driven system can be learned to good accuracy even during and after the pulse. This provides an upper bound on the minimal loss achievable by our RBM ansatz.

4- Below Eq. 5 it would be helpful to give the formulas for the quantum Fisher matrix and the energy gradient.

We have moved the corresponding formulas from Appendix B to the main text below Eq. (5) as requested.

5- On page 6 the authors discuss the step size used and mention that smaller step sizes were tested. What was the smallest step size tested and why wasn’t an integrator with adaptive step size used?

Based on this question as well as Referee 1’s interest in the time-step dependence of the full summation results (Figs. 4 and 5 in the revised version), we have performed further analysis of the time-step dependence and expanded on our observations in this regard in the manuscript.

The reduction of the time step indeed has a stabilizing effect on the dynamics both in the full summation case (Fig. 5 of the revised manuscript) and in our t-VMC calculations, although the effect is less clear in the regime of increased sampling complexity around the end of the pulse. Overall, both reducing stochastic error (by increasing the sample size or improving sampling efficiency) and taking smaller time steps does improve the stability of the simulation and reduces the likelihood of the jump instabilities. We have updated our Abstract as well as parts of the main text in order to highlight this important fact. It is remarkable, however, that an appropriate choice of the regularization scheme and hyperparameters has a similarly stabilizing effect. In contrast to reduced time step or increased sample size, this comes without an increase in computational cost. For this reason, we focus on the influence and tuning of the regularization in this work. We have expanded the Introduction to make this motivation in the Introduction.

We have chosen to present fixed-step size results in this work. It is true that for practical simulations, adaptive step-size methods can be useful as a means to improve performance by allowing the integrator to chose small time steps only in regions where it is necessary and we are aware that this has been successfully used with NQS in the literature [44]. However, we have found that it is still necessary to tune the tolerance parameters of the method in order to find an optimal balance between stability and performance. In the context of this work, we therefore focus on conceptually simpler fixed step size schemes in order to focus on our discussion of regularization effects, leaving the study of adaptive step size schemes in the presence of Monte Carlo noise to potential future work.

6- Axis labels in Fig. 9 seem misleading. The labels seem to denote a ratio, but after thinking about it for a while I believe that the “/“ is rather supposed to denote an “or”. This should be fixed.

We thank the Referee for pointing out this oversight. It is correct that in the y axis label of Figs. 9 and 10 “/” is used to denote “or” whereas the same notation is used in Fig. 11 to denote a ratio. We have changed the y axis labels of Figs. 9 and 10 to just r² to address this issue.

7- The authors refer to the quantity “1-fidelity” as the “inverse fidelity”. I think the more commonly used term is “infidelity”. The authors might consider to change this.

We agree with the Referee that “infidelity” is the more commonly used term and have updated our manuscript as suggested.

(Unless stated otherwise, numbers of equation and literature references in our reply correspond to those in the first version of our manuscript.)

---

## Editorial Decision

resubmitted